# Serum Immunoglobulin G (IgG) Subclasses in a Cohort of Systemic Sclerosis Patients

**DOI:** 10.3390/jpm13020309

**Published:** 2023-02-10

**Authors:** Chiara Pellicano, Amalia Colalillo, Giuseppina Cusano, Andrea Palladino, Marica Pellegrini, Cinzia Anna Maria Callà, Giorgia Mazzuccato, Valeria Carnazzo, Stefano Pignalosa, Luigi Di Biase, Mariapaola Marino, Umberto Basile, Edoardo Rosato

**Affiliations:** 1Department of Translational and Precision Medicine, Sapienza University of Rome, Viale dell’Università 37, 00185 Rome, Italy; 2Dipartimento di Scienze Laboratoristiche ed Infettivologiche, Fondazione Policlinico Universitario “A. Gemelli” IRCCS, Università Cattolica del Sacro Cuore, 00168 Rome, Italy; 3Dipartimento di Patologia Clinica, Ospedale Santa Maria Goretti, AUSL Latina, 04100 Latina, Italy; 4Dipartimento di Medicina e Chirurgia Traslazionale, Fondazione Policlinico Universitario “A. Gemelli” IRCCS, Università Cattolica del Sacro Cuore, 00168 Rome, Italy

**Keywords:** systemic sclerosis, IgG subclasses, ILD

## Abstract

Objectives: To assess serum immunoglobulin G (IgG) subclasses in a cohort of systemic sclerosis (SSc) patients and to evaluate the influence of IgG subclasses in the main complications of the disease. Methods: The serum level of IgG subclasses was evaluated in 67 SSc patients and 48 healthy controls (HC), matched for sex and age. Serum samples were collected and measured IgG1–4 subclasses by turbidimetry. Results: SSc patients had lower median total IgG [9.88 g/l (IQR 8.18–11.42 g/l) vs. 12.09 g/l (IQR 10.24–13.54 g/l), *p* < 0.001], IgG1 [5.09 g/l (IQR 4.25–6.38 g/l) vs. 6.03 g/l (IQR 5.39–7.90 g/l), *p* < 0.001], and IgG3 [0.59 g/l (IQR 0.40–0.77 g/l) vs. 0.80 g/l (IQR 0.46–1 g/l), *p* < 0.05] serum levels compared to HC. The logistic regression analysis showed IgG3 as the only variable associated with the diffusing capacity of the lung for carbon monoxide (DLco) ≤60% of the predicted [OR 9.734 (CI 95%: 1.312–72.221), *p* < 0.05] and modified Rodnan skin score (mRSS) [OR 1.124 (CI 95%: 1.019–1.240), *p* < 0.05], anti-topoisomerase I [OR 0.060 (CI 95%: 0.007–0.535), *p* < 0.05], and IgG3 [OR 14.062 (CI 95%: 1.352–146.229), *p* < 0.05] as variables associated with radiological interstitial lung disease (ILD). Conclusion: SSc patients have reduced levels of total IgG and an altered IgG subclass distribution compared to HC. Moreover, SSc patients show different serum IgG subclasses profiles according to the main involvement of the disease.

## 1. Introduction

Systemic sclerosis (SSc) is a systemic autoimmune disease characterized by endothelial dysfunction, skin and internal organ fibrosis, and immune system activation with autoantibodies production [1]. The pathogenesis of SSc is complex, and there are still several needs to be clarified. B cells play a critical role in systemic autoimmunity and disease expression, producing inflammatory and profibrotic cytokines [2]. Moreover, due to chronic immune stimulation, in SSc patients, there is an increased frequency of CD21^low^ B cells [3]. These anergic and exhausted B cells play the role of antigen-presenting cells (APCs) [3]. B cell activation causes polyclonal synthesis of immunoglobulins (Ig) and autoantibody production with the overproduction of free light chains (FLCs) in the serum and urine of SSc patients [4]. Autoantibodies detected in the serum of SSc patients are mainly IgG and may modulate monocyte activity towards a proinflammatory and profibrotic phenotype driving the progression of SSc [5,6]. Komura et al. [7] demonstrated increased serum IgG levels in SSc patients with interstitial lung disease (ILD) compared to SSc patients without ILD, both in SSc patients with anti-topoisomerase I antibody and in SSc patients with anti-centromere antibody. Moreover, serum IgG levels negatively correlated with the percentage of forced vital capacity (FVC) and the percentage of diffusing capacity of the lung for carbon monoxide (DLco) in SSc patients [7]. There are four subclasses of IgG termed IgG1, IgG2, IgG3, and IgG4, with different functions, although they show more than 90% of homology in the amino acid sequence [8]. Patients with different autoimmune diseases displayed different serum IgG subclass distribution compared to healthy controls (HC) [9]. Due to the lack of correlation with antibody deposition in tissues, by now, it is not possible to hypothesize a direct relationship between serum IgG subclasses level and the disease process [9]. The nature of antigens and the duration of antigen exposure may drive and regulate the production of each IgG subclass; moreover, the outcome of immune-mediated and autoimmune diseases may be influenced by IgG subclass distribution since each IgG isotype displays different biological and functional properties [8]. Although IgG subclasses could contribute to the pathogenesis of autoimmune diseases by regulating immunoglobulin, FcγR, and complement interactions, by now, few studies have analyzed serum IgG subclass levels in SSc [9]. The aims of this study were to assess serum IgG subclasses in a cohort of SSc patients and to evaluate the influence of IgG subclasses in the main complications of the disease.

## 2. Materials and Methods

### 2.1. Subjects

Sixty-seven consecutive SSc patients, fulfilling the American College of Rheumatology/European League Against Rheumatism Collaborative Criteria (2013 ACR/EULAR) for SSc [10], were enrolled in this study. We also enrolled 48 HC, matched for sex and age, and recruited among healthcare workers. Cardiopulmonary diseases not related to SSc, pulmonary arterial hypertension (PAH), a history of infection in the last 3 months, malignancies, allergic diseases, and other autoimmune diseases were the exclusion criteria. We also excluded smokers, pregnant or breastfeeding women, and patients treated in the last 6 months with immunosuppressive agents and corticosteroids at an equivalent dose of prednisone ≥10 mg/day. 

Written informed consent was obtained from all subjects enrolled in the study. The study was conducted according to the Declaration of Helsinki, and it was approved by the ethics committee of Sapienza University (IRB 0304).

### 2.2. Clinical Assessment

The modified Rodnan skin score (mRSS) was used to assess skin involvement, and the disease subset (diffuse cutaneous SSc, dcSSc or limited cutaneous SSc, lcSSc) was defined according to LeRoy et al. [11]. Disease activity and disease severity were measured by the disease activity index (DAI) [12] and the disease severity scale (DSS) [13], respectively. DAI consists of six variables with different weights: Δ-skin = 1.5, mRss > 18 = 1.5, digital ulcers (DUs) = 1.5, tendon friction rubs (TFRs) = 2.25, C-reactive protein (CRP) > 1 mg/dL = 2.25 and DLco % predicted <70% = 1.0. A cut-off ≥2.5 identified the patients with active disease [12]. DSS is a score that evaluates the null = 0, intermediate = 1, moderate = 2, severe = 3, or end-stage = 4 involvement of the general state, peripheral vessels, skin, joints/tendons, muscles, gastrointestinal tract, lungs, heart, and kidneys. DSS is useful for assessing disease severity status in individual patients both at one point in time and longitudinally since the higher the score, the greater the organ involvement and, therefore, the greater the disease severity [13].

Nailfold videocapillaroscopy (NVC) was performed to evaluate the morphology of the nailfold dermal papillary capillaries using a videocapillaroscope equipped with a 500× magnification lens (Pinnacle Studio Version 8 software, Corel, Ottawa, ON, Canada). The capillaroscopic images were classified according to the patterns of early, active, and late, according to Cutolo et al. [14]. DUs were classified according to Amanzi et al. [15].

### 2.3. Pulmonary Function Test

Pulmonary function test (PFT) parameters [forced expiratory volume in the 1st second (FEV1), FVC] and single-breath DLco were recorded by the same blinded experienced operator with a Quark PFT 2 spirometer (Cosmed, Rome, Italy) and expressed according to the standards recommended by the American/European Respiratory Society [16]. PFTs were performed at the enrollment in this study.

### 2.4. Chest High-Resolution Computed Tomography

The chest high-resolution computed tomography (HRCT) performed during the study or in the previous six months was evaluated by the same expert radiologist; radiological patterns (normal, ground glass, reticular, and honeycombing) were measured and expressed as a percentage of total lung parenchyma through Computer-Aided Lung Informatics for Pathology Evaluation and Ratings (CALIPER) software [17]. ILD was defined as fibrotic changes affecting at least 10% of the lung parenchyma, according to a previous study [18].

### 2.5. Laboratory Assessment

Samples were obtained by standard centrifugation, divided into aliquots, and stored at a temperature of −80 °C until analysis. Specimens were thawed only once and immediately assayed in a blinded fashion and in a single batch.

The four IgG subclasses concentrations were measured by turbidimetry through the employment of Human IgG and IgG subclass liquid reagent kits (The Binding Site, Birmingham, UK) with Optilite instrument according to the manufacturer’s recommendations. These assays are intended for quantifying human IgG and IgG subclasses liquid reagent kits (The Binding Site, Birmingham, UK) with technology that allows the production of highly specific antisera ensuring reliable measurement of immunoglobulins and immunoglobulin subclasses.

Concentrations are automatically calculated by reference to a standard curve stored within the instrument. The normal range for total IgG was 7–16 g/l. The normal range for subclasses is as follows: 3.82–9.29 g/l for IgG1; 2.42–7.0 g/l for IgG2; 0.22–1.76 g/l for IgG3; 0.04–0.86 g/l for IgG4.

Immunoglobulin assays are the only immunoglobulin assays calibrated to the most current standard, DA470K, to ensure accurate quantification.

Samples were tested according to the manufacturer’s instructions and serum dilutions, where necessary, were performed according to the manufacturer’s recommendations.

All the determinations were performed by an operator without knowledge of the clinical information of the handled sample. Each sample was tested twice to minimize eventual discrepancies, and all tests were performed in the same laboratory with the same instruments.

All safety precautions were taken while handling potentially harmful samples.

### 2.6. Statistical Analysis

SPSS version 26.0 software (Bioz, Los Altos, CA, USA) was used for statistical analysis. The normal distribution of data was evaluated by the Shapiro–Wilk test. Continuous variables were expressed as the median and interquartile range (IQR), and categorical variables were expressed as absolute frequencies and percentages. Student’s or Mann–Whitney’s *t*-test was performed to evaluate differences between groups. The differences between categorical variables were evaluated by the chi-square or Fisher’s exact test, as appropriate. The Spearman correlation test was used for bivariate correlations. A stepwise logistic regression analysis was used to evaluate the association between a dependent dichotomic variable (ILD or DLco ≤60% of the predicted) and continuous [age (years), duration disease (years), mRSS, IgG3 (g/l)] or cathergorical [Scl70 (yes or no)] independent variables. Results were expressed as odds ratio (OR) and 95% confidence interval (95% CI). A *p*-value < 0.05 was considered significant.

## 3. Results

Fifty-seven (85.1%) patients and 39 (81.3%) HC were females (*p* > 0.05). The median age of SSc patients and HC was similar [55 years (IQR 45–59 years) vs. 51 years (IQR 44–57 years), *p* > 0.05].

Thirty-eight (56.7%) patients were dcSSc, and 29 (43.3%) were lcSSc. The median duration of the disease was 10 years (IQR 6–15 years), and the median mRSS was 12 (IQR 7–17). Thirty (44.8%) patients were positive for anti-topoisomerase I antibodies, and 13 (19.4%) for anti-centromere; only 2 (3%) patients were RNApolymerase III positive. Median DAI and DSS were 1.67 (0.84–3.75) and 6 (4–8), respectively. Eleven (16.4%) patients had DLco ≤60% of predicted, and 16 (23.9%) patients had radiological ILD. Demographic and clinical features of SSc patients are summarized in Table 1.

### 3.1. Serum IgG and Subclasses—SSc Patients vs. HC

Seven (10.4%) SSc patients had total IgG below the normal range, while HC had total IgG >7 g/l (*p* < 0.05). Moreover, 10 (14.9%) SSc patients had IgG1 <3.82 g/l, while no HC had IgG1 below the normal range (*p* < 0.05). IgG2 were <2.42 g/l in 12 (17.9%) SSc patients and 7 (14.6%) HC (*p* > 0.05). IgG3 were below the normal range in only one (1.5%) SSc patient and in three (6.2%) HC (*p* > 0.05). IgG4 were <0.04 g/l in four (6%) SSc patients and two (4.2%) HC (*p* > 0.05). SSc patients had a statistically significant lower median value of IgG than HC [9.88 g/l (IQR 8.18–11.42 g/l) vs. 12.09 g/l (IQR 10.24–13.54 g/l), *p* < 0.001]. The median IgG1 [5.09 g/l (IQR 4.25–6.38 g/l) vs. 6.03 g/l (IQR 5.39–7.90 g/l), *p*< 0.001] and IgG3 [0.59 g/l (IQR 0.40–0.77 g/l) vs. 0.80 g/l (IQR 0.46–1 g/l), *p* < 0.05] serum levels were significantly lower in SSc patients compared to HC. The comparative analysis of the median serum IgG subclass level between SSc patients and HC is displayed in Table 2.

### 3.2. Serum IgG and Subclasses—Demographics

We found a slightly statistically significant negative linear correlation between the ages of SSc patients and serums IgG (r = −0.326, *p* < 0.01) or IgG1 (r = −0.257, *p* < 0.05). We did not find any statistically significant correlation between the ages of SSc patients and IgG2 (r = −0.217, *p* < 0.05), IgG3 (r = −0.141, *p* > 0.05), or IgG4 (r = 0.078, *p* > 0.05). The median serum levels of total IgG [10.04 g/l (IQR 8.8–11.42 g/l) vs. 9.21 g/l (IQR 6.6–9.94 g/l), *p* > 0.05], IgG1 [5.09 g/l (IQR 4.29–6.38 g/l) vs. 5.04 g/l (IQR 3.79–5.57 g/l), *p* > 0.05], IgG2 [3.66 g/l (IQR 3.05–4.64 g/l) vs. 2.57 g/l (IQR 1.98–3.77 g/l), *p* > 0.05], IgG3 [0.58 g/l (IQR 0.41–0.67 g/l) vs. 0.77 g/l (IQR 0.28–1.19 g/l), *p* > 0.05], and IgG4 [0.23 g/l (IQR 0.15–0.44 g/l) vs. 0.43 g/l (IQR 0.07–0.65 g/l), *p* > 0.05] were similar between female and male SSc patients (Table 3). 

### 3.3. Serum IgG and Subclasses—Cutaneous and Articular Involvement

All dcSSc patients had statistically significant higher levels of IgG2 compared to the lcSSc patients [3.83 g/l (IQR 3.13–5.08 g/l) vs. 3.10 g/l (IQR 2.41–3.87 g/l), *p* < 0.05], while total IgG [10.22 g/l (IQR 8.99–11.65 g/l) vs. 9.12 g/l (IQR 7.71–10.43 g/l), *p* > 0.05], IgG1 [5.10 g/l (IQR 4.30–6.38 g/l) vs. 4.97 g/l (IQR 4.10–6.02 g/l), *p* > 0.05], IgG3 [0.60 g/l (IQR 0.43–0.77 g/l) vs. 0.53 g/l (IQR 0.39–0.82 g/l), *p* > 0.05], and IgG4 [0.28 g/l (IQR 0.11–0.49 g/l) vs. 0.19 g/l (IQR 0.15–0.44 g/l), *p* > 0.05] were similar between these groups (Table 4). 

We did not find any significant correlation between disease duration or mRSS and serum levels of total IgG or of IgG subclasses (data not shown). 

Eight (11.9%) patients had TFRs, and the median finger-to-palm (FTP) distance was 0 cm (IQR 0–0.4 cm). SSc patients with TFRs had significantly higher median IgG2 compared to SSc patients without TFRs [5.45 g/l (IQR 3.41–6.60 g/l) vs. 3.50 g/l (IQR 2.74–4.24 g/l), *p* < 0.05]. We did not find any other significant differences in median serum levels of total IgG or of IgG subclasses stratifying SSc patients according to TFRs (data not shown). A slightly significant positive linear correlation exists between FTP and only the IgG2 serum level (r = 0.278, *p* < 0.05).

### 3.4. Serum IgG and Subclasses—Serology

Moreover, no statistically significant differences were found in median serum levels of total IgG or of IgG subclasses stratifying SSc patients according to autoantibody positivity (Table 5).

### 3.5. Serum IgG and Subclasses—Microvascular Involvement

SSc patients with active NVC pattern had statistically significant higher IgG4 serum levels compared to SSc patients with early NVC pattern [0.37 g/l (IQR 0.20–0.52 g/l) vs. 0.16 g/l (IQR 0.05–0.32 g/l), *p* < 0.05]. We did not find any other significant differences in the median serum levels of total IgG or IgG subclasses stratifying SSc patients according to the NVC pattern.

Eleven (16.4%) SSc patients had active DUs at the enrollment; however, we did not find any significant differences in the median serum levels of total IgG or IgG subclasses between SSc patients with DUs and SSc patients without DUs (data not shown).

### 3.6. Serum IgG and Subclasses—DAI and DSS

DAI did not correlate with total IgG (r = −0.055, *p* > 0.05), IgG1 (r = −0.124, *p* > 0.05), IgG2 (r = 0.059, *p* > 0.05), IgG3 (r = −0.033, *p* > 0.05), and IgG4 (r = 0.039, *p* > 0.05). Moreover, DSS did not correlate with total IgG (r = 0.075, *p* > 0.05), IgG1 (r = −0.088, *p* > 0.05), IgG2 (r = 0.246, *p* < 0.05), IgG3 (r = −0.037, *p* > 0.05), and IgG4 (r = −0.025, *p* > 0.05).

### 3.7. Serum IgG and Subclasses—Cardiopulmonary Involvement

Moreover, we did not find any significant correlation between systolic pulmonary arterial pressure (sPAP) and total IgG or IgG subclasses (data not shown).

Eleven (16.4%) SSc patients had DLco ≤60% of predicted, and median serum IgG3 levels were significantly higher in SSc patients with reduced DLco than SSc patients with normal DLco [0.68 g/l (IQR 0.56–1.22 g/l) vs. 0.55 g/l (IQR 0.39–0.77 g/l), *p* < 0.05]; moreover, a slightly significant negative linear correlation exists between DLco and IgG3 (r = −0.382, *p* < 0.001). We did not find any other significant differences in the median serum levels of total IgG or of IgG subclasses between patients with reduced DLco and patients with normal DLco, nor any other significant correlation between DLco and total IgG or IgG subclasses (data not shown). SSc patients with ILD had statistically significantly higher serum levels of IgG3 compared with the SSc patients without ILD [0.65 g/l (IQR 0.61–0.83 g/l) vs. 0.53 g/l (IQR 0.39–0.72 g/l), *p* < 0.05], and there was a slightly significant positive correlation between the percentage of ILD, assessed by CALIPER software at HRTC, and serum IgG3 (r = 0.305, *p* < 0.05). We did not find any other significant differences in the median serum levels of total IgG or of IgG subclasses between patients with ILD and patients without ILD, nor any other significant correlation between the percentage of ILD and total IgG or IgG subclasses (data not shown).

The logistic regression analysis showed IgG3 as the only variable associated with DLco ≤60% of the predicted [OR 9.734 (CI 95%: 1.312–72.221), *p* < 0.05] (Table 6). The logistic regression analysis showed mRSS [OR 1.124 (CI 95%: 1.019–1.240), *p* < 0.05], anti-topoisomerase I [OR 0.060 (CI 95%: 0.007–0.535), *p* < 0.05] and IgG3 [OR 14.062 (CI 95%: 1.352–146.229), *p* < 0.05] as variables associated with radiological ILD (Table 6).

## 4. Discussion

The results of our study show that SSc patients have reduced levels of total IgG compared to HC and an altered IgG subclass distribution. Moreover, SSc patients show different serum IgG subclasses profiles according to the main involvement of the disease. Interestingly, IgG3 serum levels are associated with a reduction in DLco and with ILD in SSc patients.

Several reports have described IgG subclass distribution in autoimmune diseases [7,8,9,19,20,21,22]. Each IgG isotype displays different biological and functional properties, which may influence the pathogenesis or outcome of autoimmune diseases. Moreover, autoantigens mostly stimulate the production of IgG1 or IgG3 autoantibodies because they are mainly protein antigens (T cell-dependent antigens), while few autoantibodies are polysaccharide antigens (T cell-independent antigens) that can stimulate IgG2-antoantibodies [8,23].

Zhang et al. [9] found similar serum IgG total levels between SSc patients and HC; moreover, they found increased IgG1 and IgG3 and reduced IgG2 and IgG4 in SSc patients compared to HC. In contrast with the data in our report, SSc patients have reduced levels of total IgG compared to HC, and median IgG1 and IgG3 serum levels were significantly lower in SSc patients compared to HC, while there was no difference in IgG2 and IgG4 serum levels between these groups. This might be explained by differences in patient selection, and further research is needed to resolve these discrepancies. Moreover, it could be due to a difference in disease severity and pulmonary involvement that is different between the two studies. It is mandatory to perform a multicentric study to confirm our results.

Devey et al. [24] demonstrated that the level of IgG2 anti-dsDNA antibodies correlated with joint and skin disease alone in patients affected with systemic lupus erythematosus (SLE). In psoriatic arthritis (PsA), although it has not established a correlation with any characteristics of the disease, it demonstrated an elevation of serum IgG2 levels [22]. For the first time, to our knowledge, in this study, we demonstrated that dcSSc patients have a statistically significant higher level of IgG2 compared to lcSSc patients and that a slightly positive correlation exists between IgG2 serum level and FTP. Moreover, SSc patients with TFRs have significantly higher median IgG2 compared to SSc patients without TFRs. TFRs are strongly associated with the development of articular involvement, while FTP is influenced by skin stiffness, tendon fibrosis, and arthropathies in SSc patients [25]. IgG2 switch is stimulated mainly by bacterial polysaccharides; therefore, we may hypothesize a possible role of polysaccharides antigen in the initiation or progression of cutaneous and articular involvement in SSc patients.

A recent analysis of integrating proteomics and transcriptomics demonstrated that IgG from SSc patients could induce extracellular matrix remodeling and activate fibroblast profiles [26]. Moreover, Silver et al. [27] found increased IgG levels in bronchoalveolar lavage fluid of SSc patients with ILD, and Komura et al. [7] demonstrated increased serum IgG levels in SSc patients with ILD compared to SSc patients without ILD, both in SSc patients with anti-topoisomerase I antibody and in SSc patients with anti-centromere antibody. Moreover, serum IgG levels negatively correlated with the percentage of FVC and the percentage of DLco in SSc patients [7]. This is the first study that demonstrated a significant negative linear correlation between DLco and IgG3 and a significant positive correlation between the percentage of ILD, assessed by CALIPER software at HRTC, and serum IgG3. Moreover, in our cohort, median serum IgG3 levels were significantly higher in SSc patients with reduced DLco and ILD than in SSc patients with normal DLco or without ILD. These results are confirmed by the logistic regression analysis that shows IgG3 as the only variable associated with DLco ≤60% of the predicted and mRSS, anti-topoisomerase I, and IgG3 as variables associated with radiological ILD. Although we have already hypothesized a possible role of IgG3 in the pathogenesis of primary biliary cirrhosis (PBC) [9] or mixed cryoglobulinemia (MC) [28], strengthening the hypothesis of a pathogenic role of IgG3 in liver-related autoimmunity, to date there are no studies in the literature about the possible role of IgG3 in initiation or progression of ILD in SSc patients. IgG3 subclass represents 4–8% of total IgG in the serum that are potent proinflammatory antibodies; in the course of the viral infections, there is first an increase of IgG3, followed by IgG1 [8]. Numerous infectious agents have been proposed as possible triggering factors of SSc, as of other autoimmune diseases, and the homology between viruses and autoantibody targets suggested that molecular mimicry may have a role in initiating autoimmunity [29]. IgG3, in association with IgG1, are the predominant subclasses involved in the T-cell-mediated response to protein antigens [8]. It is well known that there is an oligoclonal expansion of T-cells in the lungs and blood of SSc patients driven by chronic and specific responses to antigens [30], and EBV or CMV infections have been shown to be environmental risk factors for SSc [31], strengthening the hypothesis that molecular mimicry between chronic viral antigens and human autoantigens could be a potential driver for autoimmunity [29]. In a recent pilot study, Servaas et al. [32] confirmed the antigen-driven expansion of CD4+/CD8+ T-cells in SSc patients, identifying that clusters of T-cell clones are highly persistent over time; moreover, the authors have shown that this persistence is a result of antigenic selection. In this milieu, the observations of our study further strengthen the hypothesis that serum IgG subclass distribution might have peculiar characteristics. Indeed, in SSc patients, ILD might be the result of chronic inflammation due to a possibly chronic viral infection that stimulates the IgG3 secretion. Moreover, the presence of specific autoantigens in the lungs of SSc patients might exacerbate the IgG3 production due to the molecular mimicry mechanism.

This study has some limitations as the monocentric design and the small sample size, which cannot allow better stratification of SSc patients and prevent further analysis. Further studies are needed to confirm these preliminary results. Moreover, prospective studies with higher sample sizes and with a characterization of selective specificities among IgG subclasses to self-components are needed to evaluate the role of IgG subclasses in the main complications of SSc.

In conclusion, our cohort of SSc patients has reduced levels of total IgG and an altered IgG subclass distribution compared to HC. Moreover, SSc patients show different serum IgG subclasses profiles according to the main involvement of the disease. The concentration of subclasses IgG and their possible partial immunosuppression in individuals could be predisposing conditions of autoimmunity.

## Figures and Tables

**Table 1 jpm-13-00309-t001:** Demographic and clinical features of systemic sclerosis (SSc) patients.

Age, years, median and IQR	55 (45–59)
Female, n (%)	57 (85.1)
dcSSc, n (%)	38 (56.7)
Disease duration, years, median, and IQR	10 (6–15)
mRSS, median and IQR	12 (7–17)
Autoantibodies	
Anti-topoisomerase I, n (%)	30 (44.8)
Anti-centromere, n (%)	13 (19.4)
Anti-RNApolymerase III, n (%)	2 (3)
None, n (%)	22 (32.8)
NVC	
Early, n (%)	12 (17.9)
Active, n (%)	24 (35.8)
Late, n (%)	31 (46.3)
DAI, median and IQR	1.67 (0.84–3.75)
DSS, median and IQR	6 (4–8)
Digital ulcers, n (%)	11 (16.4)
sPAP, mmHg, median and IQR	28 (25–30)
DLco, % of predicted, median and IQR	77 (65–84)
DLco ≤ 60% of predicted, n (%)	11 (16.4)
ILD, n (%)	16 (23.9)

SSc: Systemic Sclerosis; dcSSc: Diffuse Cutaneous Systemic Sclerosis; mRSS: Modified Rodnan Skin Score; ANA: Antinuclear Antibodies; NVC: Nailfold Videocapillaroscopy; DAI: Disease Activity Index; DSS: Disease Severity Scale; sPAP: Pulmonary Arterial Systolic Pressure; DLco: Diffusing Capacity of Lung for Carbon Monoxide; ILD: Interstitial Lung Disease; IQR: Interquartile Range.

**Table 2 jpm-13-00309-t002:** Comparative analysis of serum immunoglobulin G (IgG) levels between 67 systemic sclerosis patients (SSc) and 48 healthy controls (HC). In bold are reported significant differences between groups.

	SSc	HC	*p*
IgG, g/l, median and IQR	**9.88 (8.18–11.42)**	**12.09 (10.24–13.54)**	**<0.001**
IgG1, g/l, median and IQR	**5.09 (4.25–6.38)**	**6.03 (5.39–7.90)**	**<0.001**
IgG2, g/l, median and IQR	3.58 (2.87–4.64)	4 (2.62–4.90)	>0.05
IgG3, g/l, median and IQR	**0.59 (0.40–0.77)**	**0.80 (0.46–1)**	**<0.05**
IgG4, g/l, median and IQR	0.24 (0.12–0.49)	0.32 (0.20–0.51)	>0.05
IgG1/IgG, %, median and IQR	0.53 (0.48–0.61)	0.56 (0.50–0.60)	>0.05
IgG2/IgG, %, median and IQR	0.38 (0.30–0.43)	0.35 (0.27–0.40)	>0.05
IgG3/IgG, %, median and IQR	0.06 (0.04–0.08)	0.06 (0.04–0.08)	>0.05
IgG4/IgG, %, median and IQR	0.03 (0.01–0.05)	0.03 (0.02–0.05)	>0.05

SSc: Systemic Sclerosis; HC: healthy control; IQR: Interquartile Range.

**Table 3 jpm-13-00309-t003:** Comparative analysis of serum immunoglobulin G (IgG) levels in systemic sclerosis patients (SSc) according to sex.

	Female SSc (n = 57)	Male SSc (n = 10)	*p*
IgG, g/l, median and IQR	10.04 (8.8–11.42)	9.21 (6.6–9.94)	>0.05
IgG1, g/l, median and IQR	5.09 (4.29–6.38)	5.04 (3.79–5.57)	>0.05
IgG2, g/l, median and IQR	3.66 (3.05–4.64)	2.57 (1.98–3.77)	>0.05
IgG3, g/l, median and IQR	0.58 (0.41–0.67)	0.77 (0.28–1.19)	>0.05
IgG4, g/l, median and IQR	0.23 (0.15–0.44)	0.43 (0.07–0.65)	>0.05

SSc: systemic sclerosis; IQR: Interquartile Range.

**Table 4 jpm-13-00309-t004:** Comparative analysis of serum immunoglobulin G (IgG) levels in systemic sclerosis patients (SSc) according to disease subset. In bold are reported significant differences between groups.

	dcSSc (n = 38)	lcSSc (n = 29)	*p*
IgG, g/l, median and IQR	10.22 (8.99–11.65)	9.12 (7.71–10.43)	>0.05
IgG1, g/l, median and IQR	5.10 (4.30–6.38)	4.97 (4.10–6.02)	>0.05
IgG2, g/l, median and IQR	**3.83 (3.13–5.08)**	**3.10 (2.41–3.87)**	**<0.05**
IgG3, g/l, median and IQR	0.60 (0.43–0.77)	0.53 (0.39–0.82)	>0.05
IgG4, g/l, median and IQR	0.28 (0.11–0.49)	0.19 (0.15–0.44)	>0.05

dcSSc: diffuse cutaneous systemic sclerosis; lcSSc: limited cutaneous systemic sclerosis; IQR: Interquartile Range.

**Table 5 jpm-13-00309-t005:** Comparative analysis of serum immunoglobulin G (IgG) levels in systemic sclerosis patients (SSc) according to autoantibody positivity.

	Centromere (n = 13)	None (n = 22)	RNApolymerase III (n = 2)	Topoisomerase I (n = 30)	*p*
IgG, g/l, median and IQR	8.8 (7.48–10.04)	9.91 (8.88–11.65)	14.78 (11.28–18.28)	10.05 (8.63–11.42)	>0.05
IgG1, g/l, median and IQR	4.59 (4.1–5.51)	5.63 (4.58–6.77)	7.63 (7.1–8.17)	4.78 (4.23–5.85)	>0.05
IgG2, g/l, median and IQR	3.1 (1.91–3.87)	3.14 (2.74–4.39)	5.56 (3.32–7.8)	3.88 (3.13–4.77)	>0.05
IgG3, g/l, median and IQR	0.55 (0.3–0.63)	0.53 (0.4–1.06)	0.94 (0.63–1.25)	0.59 (0.42–0.69)	>0.05
IgG4, g/l, median and IQR	0.16 (0.15–0.39)	0.24 (0.15–0.51)	0.65 (0.24–1.06)	0.27 (0.11–0.45)	>0.05

IQR: Interquartile Range.

**Table 6 jpm-13-00309-t006:** Logistic regression analysis. In bold are reported significant variables.

	DLco ≤ 60% of Predicted		ILD	
	OR (CI 95%)	*p*	OR (CI 95%)	*p*
Age, years	1.032 (0.960–1.110)	>0.05	1.036 (0.958–1.211)	>0.05
Disease duration, years	0.965 (0.867–1.074)	>0.05	1 (0.930–1.076)	>0.05
mRSS	1.063 (0.971–1.163)	>0.05	**1.124 (1.019–1.240)**	**<0.05**
Anti-topoisomerase I	0.651 (0.115–3.666)	>0.05	**0.060 (0.007–0.535)**	**<0.05**
IgG3, g/l	**9.734 (1.312–72.221)**	**<0.05**	**14.062 (1.352–146.229)**	**<0.05**

mRSS: Modified Rodnan Skin Score; DLco: Diffusing Capacity of Lung for Carbon Monoxide; ILD: Interstitial Lung Disease; OR: Odds Ratio; CI: Confidence Interval.

## Data Availability

The original data presented in the study are included in the manuscript. Further requests should be addressed to the corresponding author.

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
