# Peer review of "Serum Immunoglobulin G (IgG) Subclasses in a Cohort of Systemic Sclerosis Patients"

_jpm, 2023, doi:10.3390/jpm13020309_

Round 1
Reviewer 1 Report
Interesting topic, although preliminary data that should be verified in larger SSc cohorts in order to validate pathogenetic value and prognostic/ red flag indicators
Introduction provides enough background based on relevant references
Material and methods
lines 92-93: please give a more detailed presentation on the activity (DAI) and severity (DSS) scores used as it is not clear which indexes and cut-off levels;
line 94: please see the definition of NVC and change "the level of distal phalanx" with a more suitable formulation
lines 123-124: no normal range for total IgG was provided
Results
lines 146-147 - it is obvious that the differences between gender and age of 2 groups are not statistically significant as HC are age and sex-matched
please make a paragraphs with comments about clinical and serological findings as well as DAI, DSS; data are only summarized in table 1.
lines 160-161; as providing data about normal range of IgG, the authors will be entitled to comment about the severity of decreased IgG in SSc and subset analysis
lines 164-165 - please add data/ table about IgG subclass distribution among dcSSc vs lcSSc
lines 160-223: please reorganize results in order to better highlight outcomes and potential red flags; e.g. ; IgG and subclases – demographics IgG and subclases – clinical findings; IgG and subclases - serology, etc
Author Response
Reply to the Reviewers
Reviewer #1
Interesting topic, although preliminary data that should be verified in larger SSc cohorts in order to validate pathogenetic value and prognostic/ red flag indicators.
Authors' Reply: We thank the reviewer for his/her comment.
Introduction provides enough background based on relevant references
Authors' Reply: We thank the reviewer for his/her comment.
Material and methods
lines 92-93: please give a more detailed presentation on the activity (DAI) and severity (DSS) scores used as it is not clear which indexes and cut-off levels;
Authors' Reply: We thank the reviewer for his/her suggestion to improve the quality of our manuscript and we gave a more detailed presentation of DAI and DSS. We added the following sentences in matherials and methods section, accordingly:
“DAI consists of six variables with different weight: Δ-skin=1.5, mRss>18=1.5, digital ulcers (DUs)=1.5, tendon friction rubs (TFRs)=2.25, C-reactive protein (CRP)>1 mg/dL=2.25 and DLco % predicted <70%=1.0. A cut-off ≥2.5 identify patients with active disease [12]”.
“DSS is based on nine elements: general state, peripheral vessels, skin, joints/tendons, muscles, gastrointestinal tract, lungs, heart and kidneys. Each organ or system is assessed separately with a score ranging from a minimum of 0 to a maximum of 4, in consideration of its null, intermediate, moderate, severe or "end-stage" involvement. DSS is useful for assessing disease severity status in individual patients both at one point in time and longitudinally since the higher the score the greater the organ involvement and therefore the grater the disease severity [13].”
line 94: please see the definition of NVC and change "the level of distal phalanx" with a more suitable formulation
Authors' Reply: We thank the reviewer for his/her suggestion to improve the quality of our manuscript and we modified accordingly.
lines 123-124: no normal range for total IgG was provided
Authors' Reply: We thank the reviewer for point out this mistake. We added normal range for total IgG.
Results
lines 146-147 - it is obvious that the differences between gender and age of 2 groups are not statistically significant as HC are age and sex-matched
Authors' Reply: We thank the reviewer for point out this. We have included these results for completeness and accuracy.
please make a paragraphs with comments about clinical and serological findings as well as DAI, DSS; data are only summarized in table 1.
Authors' Reply: We thank the reviewer for his/her suggestion to improve the quality of our manuscript and we modified results section accordingly.
lines 160-161; as providing data about normal range of IgG, the authors will be entitled to comment about the severity of decreased IgG in SSc and subset analysis
Authors' Reply: We thank the reviewer for his/her suggestion to improve the quality of our manuscript and we added these data in the results section.
lines 164-165 - please add data/ table about IgG subclass distribution among dcSSc vs lcSSc
Authors' Reply: We thank the reviewer for his/her suggestion to improve the quality of our manuscript and we added new Table 4 which display IgG subclass distribution among dcSSc vs lcSSc.
lines 160-223: please reorganize results in order to better highlight outcomes and potential red flags; e.g. ; IgG and subclases – demographics IgG and subclases – clinical findings; IgG and subclases - serology, etc
Authors' Reply: We thank the reviewer for his/her suggestion to improve the quality of our manuscript and we modified results section accordingly.
Reviewer 2 Report
The authors present interesting data regarding distribution of IgG subclasses in a relatively high group of Ssc patients. They also present interesting associations, one of the most important being the association between serum IgG3 and decreased DLCO. There are some minor aspects to be still discussed:
1. Methods section: I suppose the means of the two findings of IgG analysis were reported in the manuscript? It should be mentioned.
2. Results section:
2.1. In table 2 significant differences between groups should be highlighted (bolded)
2.2. I think that two additional tables where the authors compare the levels of IgG levels between male and female patients and also lcSsc and DSsc would make easier for the reader to see the differences instead of being in text.
2.3. Fig 1 could be deleted.
2.4. The authors should also acknowledge that all the correlations they found were rather weak and variable s that have correlation coefficients between 0-0.25 or -0.25-0 are in fact not correlated https://www.biochemia-medica.com/en/journal/17/1/10.11613/BM.2007.002/fullArticle
Also, it is not clear if authors used either Pearson or Spearman correlation coefficients, as non-parametric test should be used due to small sample size (Spearman).
2.5. In table 3 an additional column with p values according to the second logistic regression model is missing.
3. Discussion section- when comparing the results with those found by Zhang et al the authors state that “This might be explained by differences in patient selection “. However they do not provide an additional explanation? Could it be a difference in disease severity and pulmonary involvement that is different between the two studies?
Best regards and congratulations on your work!
Author Response
Reply to the Reviewers
Reviewer #2
The authors present interesting data regarding distribution of IgG subclasses in a relatively high group of Ssc patients. They also present interesting associations, one of the most important being the association between serum IgG3 and decreased DLCO.
Authors' Reply: We thank the reviewer for his/her comment.
There are some minor aspects to be still discussed:
1.Methods section: I suppose the means of the two findings of IgG analysis were reported in the manuscript? It should be mentioned.
Authors' Reply:
2.Results section:
2.1. In table 2 significant differences between groups should be highlighted (bolded)
Authors' Reply: We thank the reviewer for his/her suggestion to improve the quality of our manuscript and we modified tables accordingly.
2.2. I think that two additional tables where the authors compare the levels of IgG levels between male and female patients and also lcSsc and DSsc would make easier for the reader to see the differences instead of being in text.
Authors' Reply: We thank the reviewer for his/her suggestion to improve the quality of our manuscript and we added new Table 3 and Table 4. Moreover, we added a new Table 5 according to reviewer #1 suggestion so Table 3 become now Table 6.
2.3. Fig 1 could be deleted.
Authors' Reply: We thank the reviewer for his/her suggestion to improve the quality of our manuscript and we deleted Figure 1 accordingly.
2.4. The authors should also acknowledge that all the correlations they found were rather weak and variable s that have correlation coefficients between 0-0.25 or -0.25-0 are in fact not correlated https://www.biochemia-medica.com/en/journal/17/1/10.11613/BM.2007.002/fullArticle
Authors' Reply: We thank the reviewer for his/her comment. We are aware of the weakiness of the correlation found so we tempered the result section accordingly.
Also, it is not clear if authors used either Pearson or Spearman correlation coefficients, as non-parametric test should be used due to small sample size (Spearman).
Authors' Reply: We thank the reviewer for his/her comment. We used Spearman correlation coefficients due to data distribution significantly deviates from normal distribution in our cohort. We modified the statistical analysis paragraph accordingly.
2.5 In table 3 an additional column with p values according to the second logistic regression model is missing.
Authors' Reply: We thank the reviewer for point out this mistake. We added an additional column with p values according to the second logistic regression model, in table 3.
- Discussion section- when comparing the results with those found by Zhang et al the authors state that “This might be explained by differences in patient selection “. However they do not provide an additional explanation? Could it be a difference in disease severity and pulmonary involvement that is different between the two studies?
Authors' Reply: We thank the reviewer for his/her suggestion to improve our manuscript. We added in discussion section the following sentence “Moreover, it could be due to a difference in disease severity and pulmonary involvement that is different between the two studies. Its mandatory to perform a multicentric study to confirm our results.”